# Single-gap isotropic $s-$wave superconductivity in single crystals AuSn$_4$

Sunil Ghimire[1,2], Kamal R. Joshi[1,2], Elizabeth H. Krenkel[1,2], Makariy A. Tanatar[1,2], Marcin Kończykowski[3], Romain Grasset[3], Paul C. Canfield[1,2] and Ruslan Prozorov[1,2⋆]

**1** Ames National Laboratory, Ames, Iowa 50011, USA
**2** Department of Physics & Astronomy, Iowa State University, Ames, Iowa 50011, USA
**3** Laboratoire des Solides Irradiés, CEA/DRF/lRAMIS, École Polytechnique, CNRS, Institut Polytechnique de Paris, F-91128 Palaiseau, France

⋆ prozorov@ameslab.gov

## Abstract

London, $\lambda_L(T)$, and Campbell, $\lambda_C(T)$, penetration depths were measured in single crystals of a topological superconductor candidate AuSn$_4$. At low temperatures, $\lambda_L(T)$ is exponentially attenuated and, if fitted with the power law, $\lambda(T) \sim T^n$, gives exponents $n > 4$, indistinguishable from the isotropic single $s-$wave gap Bardeen-Cooper-Schrieffer (BCS) asymptotic. The superfluid density fits perfectly in the entire temperature range to the BCS theory. The superconducting transition temperature, $T_c = 2.40 \pm 0.05$ K, does not change after 2.5 MeV electron irradiation, indicating the validity of the Anderson theorem for isotropic $s-$wave superconductors. Campbell penetration depth before and after electron irradiation shows no hysteresis between the zero-field cooling (ZFC) and field cooling (FC) protocols, consistent with the parabolic pinning potential. Interestingly, the critical current density estimated from the original Campbell theory decreases after irradiation, implying that a more sophisticated theory involving collective effects is needed to describe vortex pinning in this system. In general, our thermodynamic measurements strongly suggest that the bulk response of the AuSn$_4$ crystals is fully consistent with the isotropic $s-$wave weak-coupling BCS superconductivity.



# 1 Introduction

In recent years, superconductors with topological features in their electronic bandstructure have attracted significant interest for various novel features predicted by a well-developed theory, for example, emerging zero-energy excitations called Majorana fermions [1]. On the material side, the search for topological superconductors (TSCs) is very active but so far has yielded only a few "candidates" whose topological properties have not yet been fully confirmed experimentally, including $UTe_2$ [2], $Sr_2RuO_4$ [3–5], $UPt_3$ [6], 2M-$WS_2$ [7], and $M_xBi_2Se_3$ with M=Cu [8,9]. The subject of this study, $AuSn_4$, is another promising TSC candidate with theoretically predicted non-trivial topological characteristics [10–13].

The superconductivity in orthorhombic $AuSn_4$ with a transition temperature to the superconducting state, $T_c = 2.4$ K, was discovered in 1962 [14]. This compound is isostructural to $PtSn_4$ [15] and $PdSn_4$ [16], which are not superconductors. The first principal study suggests semimetallic behavior with type I nodes [12]. The magneto-trasnport measurements show two-dimensional (2D) superconductivity in $AuSn_4$ [11, 17]. Recently, ARPES measurements supported by DFT calculations [13] revealed nearly degenerate polytypes in $AuSn_4$ crystals, making it a unique case of a three-dimensional (3D) electronic band structure with properties of a low-dimensional layered material. Thermodynamic magnetization and specific heat measurement in $AuSn_4$ single crystals are consistent with conventional nodeless $s-$wave Bardeen-Cooper-Schrieffer (BCS) [18, 19] superconductivity [11]. Scanning tunneling microscopy (STM) measurements determined the superconducting gap to $T_c$ ratio close to the $s-$wave BCS value of $\Delta/T_c = 1.76$ [13]. However, other STM measurements suggest unconventional 2D superconductivity with a mixture of $p-$wave surface states and $s-$wave bulk [10]. Clearly, more measurements are required for an objective and conclusive determination of the nature of superconductivity in $AuSn_4$.

Here, we probe the bulk nature of superconductivity in $AuSn_4$ single crystals by measuring London and Campbell penetration depths using a highly sensitive tunnel-diode resonator (TDR). Furthermore, we examine the response to a controlled non-magnetic point-like disorder induced by 2.5 MeV electron irradiation. We conclude that $AuSn_4$ is a robust isotropic $s-$wave superconductor in the bulk. However, we cannot exclude the possibility that it could have a different type of superconductivity in the surface atomic layers, where the STM is most sensitive.

# 2 Samples and methods

Single crystals of $AuSn_4$ were grown with excess Sn flux [13, 20, 21]. High-purity Au and Sn were mixed in a 12:88 ratio in a fritted crucible [22,23] and sealed in a quartz ampoule under an Ar gas atmosphere. The ampoule was heated to 1100 °C over 12 hours, then cooled to 250 °C in 12 hours, and significantly slower to 230 °C over 90 hours. The ampoule was held at this temperature for 48 hours prior to removal from the furnace.

The London penetration depth, $\lambda(T)$, was measured using a sensitive frequency-domain self-oscillating tunnel-diode resonator (TDR) operating at a frequency of around 14 MHz. Measurements were performed in a $^3$He cryostat with a base temperature of $\approx 400$ mK, which is $0.17T_c$, allowing us to examine the low-temperature limit, which starts below approximately $T_c/3$, below which the superconducting gap is approximately constant [19]. The experimental setup, measurement protocols, and calibration are described in detail elsewhere [24–28].

Briefly, the sample placed inside the inductor of the $LC-$ tank circuit affects the total inductance, $L$, leading to a shift of its resonant frequency, $2\pi f = 1/\sqrt{LC}$ by the amount proportional to the magnetic susceptibility of the sample, $\Delta f = G\chi(T)$, where $G$ is the cali-

bration constant [25, 27]. A small excitation magnetic field of the setup, $\sim 20$mOe, ensures the regime of a small-amplitude linear magnetic response where $\chi = \lambda_m/R \tanh(R/\lambda_m) - 1$. Here $R$ is the effective dimension of the sample and $\lambda_m$ is the total measured magnetic penetration depth [27]. The sample of this study had dimensions $0.6 \times 0.4 \times 0.1$ mm$^3$, which gives the effective dimension $R = 84.1\,\mu$m, calculated using the procedure described in Ref. [27]. With a penetration depth smaller than a few micrometers for most of the temperature range, we can simplify the relation for magnetic susceptibility to $\chi \approx \lambda_m/R - 1$. Therefore, the change in $\lambda_m(T)$ with respect to a reference point at the lowest temperature, $\lambda_m(T_{min})$ is proportional to the relative frequency shift, $\Delta\lambda_m(T) = \lambda_m(T) - \lambda_m(0.4\,\text{K}) = (R/G)\Delta f$. For all practical purposes, $\lambda_m(0) \approx \lambda_m(0.4\,\text{K})$. Although the calibration constant $G$ can be calculated, in our setup it is measured directly by mechanically extracting the sample from the coil at low temperature, thus providing a robust calibration specific for each sample studied.

In a zero magnetic field, there are no vortices and the measured penetration depth is the London penetration depth, $\lambda_L(T)$. Therefore, by measuring the change, $\Delta\lambda_L(T) = \lambda_m(T, B = 0)$, and adding the absolute value, $\lambda(0)$ determined from other measurements, such as fitting $\Delta\lambda(T)$ in Fig. 2, a full London length is obtained, $\lambda_L(T) = \lambda_L(0) + \Delta\lambda_L(T)$. In the presence of a magnetic field, in a small-amplitude linear AC response, the vortex-mediated Campbell length adds to the London length, $\lambda_m^2 = \lambda_C^2 + \lambda_L^2$ [29–31]. Upon approaching the transition temperature, $T_c$, the penetration depth can only increase up to a normal-metal skin depth. Since the magnetoresistance at $T_c$ is known and in this case is negligible, the value at $T_c$ is used as a fixed reference point [13]. Measuring $\lambda_m(T, B)$ in finite magnetic fields, the curves are shifted vertically, so that the saturation flat parts above $T_c$ match. Once such a vertical shift procedure is performed, the full Campbell penetration depth is extracted from $\lambda_C = \sqrt{\lambda_m^2 - \lambda_L^2}$.

Point-like disorder was introduced at the SIRIUS facility in the Laboratoire des Solides Irradiés at École Polytechnique in Palaiseau, France. Electrons, accelerated in a pelletron-type linear accelerator to 2.5 MeV, knock out ions, creating vacancy-interstitial Frenkel pairs [32, 33]. During irradiation, the sample is immersed in liquid hydrogen at around 20 K. This ensures efficient heat removal upon impact and prevents immediate recombination and migration of the produced atomic defects. The acquired irradiation dose is determined by measuring the total charge collected by a Faraday cage located behind the sample. As such, the acquired dose is measured in the "natural" units of C/cm$^2$, which is equal to $1\text{C/cm}^2 \equiv 1/e \approx 6.24 \times 10^{18}$ electrons per cm$^2$. Upon warming to room temperature, some defects recombine, and some migrate to various sinks (dislocations, surfaces, etc.). This leaves a metastable population, about 70%, of point-like defects [34, 35]. Importantly, the same sample has been measured before and after electron irradiation.

## 3 Results

### 3.1 London penetration depth

Figure 1 shows the low-temperature dependence of the change in the London penetration depth, $\Delta\lambda(T) = \lambda(T) - \lambda(T_{min} = 0.4\,\text{K})$ before (blue circles) and after 2.5 C/cm$^2$ electron irradiation (red circles). The upper left inset shows the exponent $n$ determined from the power-law fitting, $\Delta\lambda(T) \sim At^n$, as a function of the upper fitting limit, $t_{max} = T_{max}/T_c$. The solid lines in the main frame show an example of such a fitting with $t_{max} = 0.4$. The results show a robust and consistent behavior with $n \geq 4$, indicating experimentally indistinguishable from the exponential temperature dependence. The exponent, $n$, decreased after irradiation as it should be in an $s-$wave superconductor [36, 37].

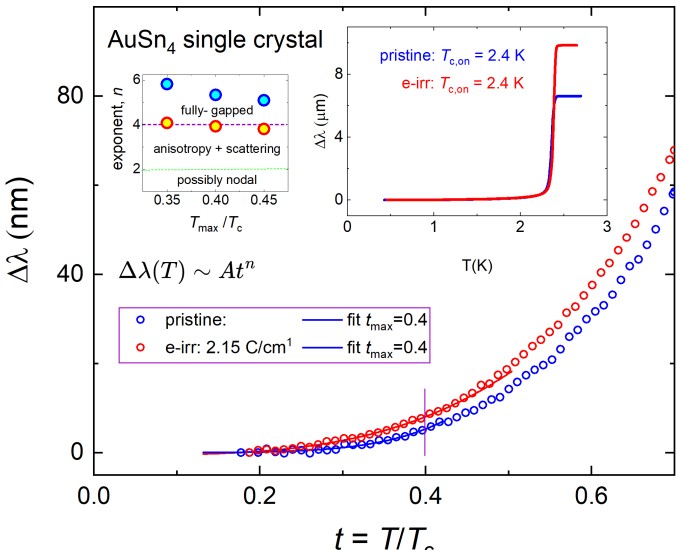

Figure 1: Main Panel: Low-temperature temperature variation of the London penetration depth $\Delta\lambda(T) = \lambda(T) - \lambda(0.4\,\mathrm{K})$ as a function of normalized temperature, $t = T/T_c$, for pristine (blue circles) and irradiated at 2.5 C/cm$^2$ (red circles) single crystal of AuSn$_4$. Lines show fits to the power law, $\Delta\lambda(T) \sim At^n$, with the upper range of $t_{max} = 0.4$. The top right inset shows the $\Delta\lambda(T)$ in the whole temperature range, showing sharp superconducting transition with onset $T_c = 2.4\,\mathrm{K}$ for both pristine and electron irradiated state. The top left inset shows the exponent $n$ versus the upper limit of the power-law fitting, $t_{max} = T_{max}/T_c$, indicating robustness of the power law, experimentally indistinguishable from exponential.

The upper right inset of Fig.1 shows $\Delta\lambda(T)$ of the same sample in its pristine state and after 2.15 C/cm$^2$ electron irradiation as a function of absolute temperature $T$. One might think that for some reason (e.g., defect annealing and recombination), there was no increase in disorder after irradiation. It is straightforward to prove that this is not the case. The saturation of the measured $\lambda(T)$ above $T_c$ occurs when it reaches the skin depth of the normal state, $\delta_{\mathrm{skin}} = \sqrt{\rho/\mu_0\pi f}$, where $\mu_0 = 4\pi \times 10^{-7}$ H/m is the vacuum permeability, and $\rho$ is the resistance. More precisely, $\delta_{\mathrm{skin}}(T_c) = 2\lambda(T_c)$ [38]. We did not measure resistivity in this particular AuSn$_4$ sample, but we directly compared resistivity from transport measurements and extracted from the skin depth on the same samples of other compounds and always found good quantitative agreement [39,40]. Furthermore, the upper critical fields are small, $H_{c2}^{\|ab} = 130$ Oe and $H_{c2}^{\|c} = 90$ Oe [11]. Consulting with published magnetoresistance [13], we find that the expected variation of $\delta_{\mathrm{skin}}$ just above $T_c$ is negligible. On the other hand, the top right inset in Fig.1 shows a substantial increase in saturation value after electron irradiation. This proves a substantial increase of resistivity, which can only be due to added disorder scattering. Therefore, the fact that the superconducting transition temperature $T_c$ remains unchanged is consistent with the Anderson theorem for isotropic $s-$wave superconductors [41,42]. We have observed similar robust superconductivity in another low$-T_c$ superconductor with non-trivial topology, LaNiGa$_2$ [43].

The exponential temperature dependence of $\lambda(T)$ can be fitted with the well-known low-temperature asymptotic BCS, $\Delta\lambda(T) = \lambda(0)\sqrt{\frac{\pi\delta}{2t}}e^{-\frac{\delta}{t}}$ [19], where the ratio $\delta = \Delta(0)/T_c$ was fixed at $\delta \approx 1.764$, leaving only one free parameter $\lambda(0)$. The fitting is shown in the top panel of Fig.2. It produces $\lambda(0) = 150$ nm in the pristine state (blue fitting curve and blue data symbols) and $\lambda(0) = 258$ nm after 2.15 C/cm$^2$ electron irradiation (red curve and symbols).

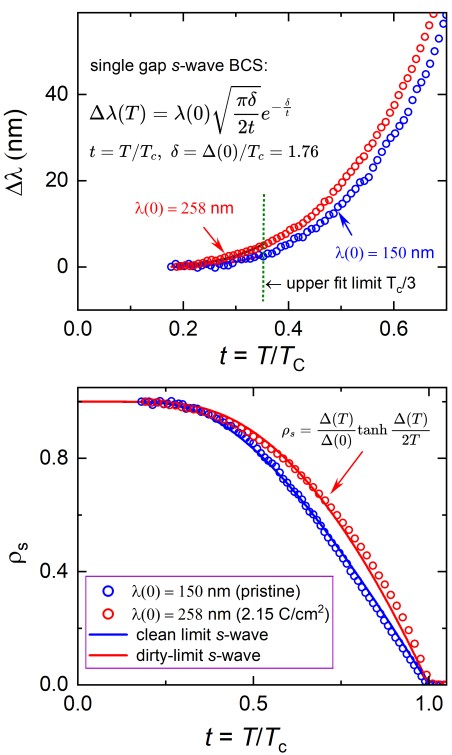

Figure 2: Top panel. Fit to the BCS low-temperature asymptotic, $\Delta\lambda(T) = \lambda(0)\sqrt{\frac{\pi\delta}{2t}}e^{-\frac{\delta}{t}}$ with a fixed ratio $\delta = \Delta(0)/T_c \approx 1.76$ leaving only one free parameter, $\lambda(0) = 150$ nm in the pristine sample (blue fitting curve and blue data symbols) and $\lambda(0) = 258$ nm after 2.15 C/cm$^2$ electron irradiation (red curve and symbols). Bottom panel: Superfluid density calculated from the data, $\rho_s(T) = (1+\Delta\lambda(T)/\lambda(0))^{-2}$. Solid lines show self-consistent full temperature range calculations using Eilenberger formalism for pristine (blue line) and irradiated (red line) states. The known analytical expression for the $s-$wave dirty limit is shown in [44].

With these numbers, we can calculate the superfluid density in the full temperature range using $\rho_s(T) \equiv (\lambda(0)/\lambda(T))^2 = (1+\Delta\lambda(T)/\lambda(0))^{-2}$. The bottom panel of Fig.2 shows $\rho_s(T)$ by blue and red circles for the pristine and irradiated states of the same sample, respectively. The theoretical lines of the clean (blue) and dirty (red) limits were calculated self-consistently using the Eilenberger formalism [45]. The analytical dirty limit formula, $\rho_s = (\Delta(T)/\Delta(0))\tanh(\Delta(T)/2T)$ reproduces the numerical calculation precisely [44]. We note that due to a limited number of data points, good fits of $\lambda(T)$ can also be obtained with slightly different ratios of $\Delta(0)/T_c$. However, then the full-range superfluid density curve does not fit. It fits only with the weak-coupling isotropic BCS value of 1.764. In summary, Fig.2 shows that the classical BCS theory describes the experimental data well.

To summarize our findings from measurements of the London penetration depth, $\lambda(T)$, several independent characteristics: (1) low-temperature behavior of $\lambda(T)$; (2) full temperature range behavior of $\rho_s$; (3) disorder-independent $T_c$ before and after electron irradiation, fully agree with the BCS theory for the isotropic $s-wave$ gap with the ratio $\delta = \Delta(0)/T_c \approx 1.76$. This is the nature of superconductivity in the bulk of AuSn$_4$ crystals. However, our measurements would not pick up a tiny signal coming from the surface atomic layers, so unconventional topological features are still possible.

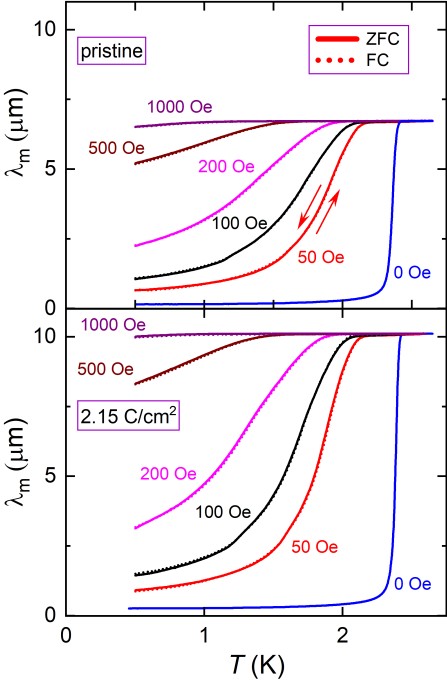

Figure 3: Temperature variation of the measured magnetic penetration depth, $\lambda_m$, before (top panel) and after (bottom panel) electron irradiation, measured with the various dc magnetic fields applied along the $c-$ axis. The field values are shown. Solid lines correspond to zero-field cooling (ZFC), and dotted lines correspond to field cooling (FC) protocols. For one curve, this is shown by arrows. The ZFC and FC curves are indistinguishable, implying that the process is completely reversible, indicating the pinning potential's parabolic shape. Note that the axes scales are the same in the top and bottom panels, aiding in a visual comparison of the effect of irradiation.

## 3.2 Campbell penetration depth

The temperature variation of the magnetic penetration depth before (top panel) and after (bottom panel) electron irradiation, measured in various dc magnetic fields applied along the $c-$ axis, is shown in Fig.3. The field values are shown next to each curve. Solid lines correspond to zero-field cooling (ZFC) in all curves, and dotted lines correspond to field cooling (FC) protocols. For one curve, this is shown by arrows. The ZFC and FC curves are indistinguishable, implying that the process is totally reversible, which indicates a parabolic shape of the pinning potential.

In the presence of an external DC magnetic field, Abrikosov vortices penetrate the sample and form a vortex lattice. Then the measured penetration depth, $\lambda_m$, has two contributions, the usual London penetration depth that in this section we explicitly denote as $\lambda_L$, and the Campbell penetration depth $\lambda_C$, which is a characteristic length scale over which a small ac perturbation is transmitted elastically by a vortex lattice into the sample [46–49]. More specifically, the amplitude of the ac perturbation must be small enough so that the vortices remain in their potential well, and their motion is described by the reversible linear elastic response. In this case, $\lambda_m^2 = \lambda_L^2 + \lambda_C^2$ [29,50]. This requirement of a very small amplitude makes most conventional ac susceptibility techniques inapplicable for the measurements of the Campbell length. Specialized frequency domain resonators with sufficient sensitivity to a small excitation ac magnetic field are needed [51,52]. Until now, only a few experimental studies have been published [31,51–54].

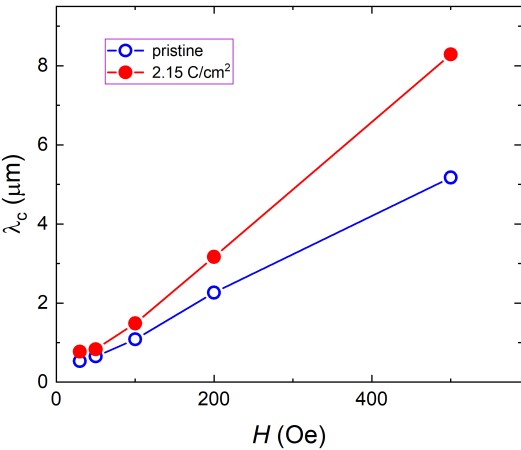

Figure 4: Campbell penetration depth, $\lambda_C^2 = \sqrt{\lambda_m^2 - \lambda_L^2}$ as a function of an applied magnetic field, $H$, evaluated from the data shown in Fig. 3 at a fixed temperature of $T = 0.5$ K, for a FC protocol comparing pristine (blue symbols) and irradiated (red symbols) states of the same sample.

Figure 3 shows the temperature-dependent variation of the magnetic penetration depth, $\lambda_m(T) = \lambda_L(0) + \Delta\lambda_m(T)$, for different values of the dc magnetic field applied parallel to the sample $c$−axis. For $\lambda_L(0)$, we have used the values obtained from the BCS fit; see the upper panel of Fig. 2. Then, we assumed that, above $T_c$, the resistivity is field independent, so we adjusted other curves to match that value. The top panel shows a pristine state, and the bottom panel shows the same sample after electron irradiation.

Generally speaking, the Campbell penetration depth can exhibit a hysteresis upon warming and cooling, indicating an anharmonic (non-parabolic) pinning potential and/or strong pinning [49, 51, 54, 55]. Therefore, there are two types of measurement protocols: zero-field cooling (ZFC) and field cooling (FC). In the ZFC protocol, the Campbell length is measured on warming after the sample was cooled in a zero magnetic field and the target field was applied at the base temperature (solid lines in Fig. 3). In the FC protocol, measurements are performed on cooling in a target magnetic field applied above $T_c$ (dotted lines in Fig. 3). For both pristine and irradiated states, $\lambda_m(T)$ shows a monotonic increase with temperature, and there is no hysteresis between the ZFC and FC protocols. To aid in visualizing the effect of irradiation, the scales of the axes in Fig. 3 are the same in the top and bottom panels. It is clear that the measured penetration depth has increased after electron irradiation.

Figure 4 shows the Campbell penetration depth as a function of an applied magnetic field, $H$, evaluated from the data shown in Fig. 3 at a fixed temperature of $T = 0.5$ K for a FC protocol comparing pristine (blue symbols) and irradiated (red symbols) states of the same sample. The Campbell length $\lambda_C$ increases after electron irradiation. In the simple Campbell model [46, 47], $\lambda_C^2 = \phi_0 H/\alpha$, where $\phi_0$ is the magnetic flux quantum and $\alpha$ is the curvature of the pinning potential, $\alpha = d^2U/dr^2$. The critical current density $j_c = \alpha r_p/\phi_0 = Hr_p/\lambda_C^2$, where $r_p$ is the radius of the pinning potential, usually assumed to be of the order of the coherence length, $\xi$. We note that this critical current is not the same as the persistent current obtained in conventional magnetization measurements, which is based on the Bean model that assumes a constant vortex density gradient [56, 57]. In the present measurements, the critical current is a parameter of the model defining the equilibrium Campbell length without persistent Bean currents present. It represents a theoretical current density supported by a specific pining potential, $U(r)$. The conventional measured current density is lower due to magnetic relaxation, which is very fast on short time scales and later slows down to become time-logarithmic [58].

In a more general picture, $\alpha$ is determined by the elementary pinning forces [48, 49, 59]. In the original model with a fixed $r_p$, the Campbell length is expected to scale as $\lambda_C \sim \sqrt{H}$, but Fig.4 shows a practically linear temperature dependence, especially after irradiation. This indicates that vortex pinning in $AuSn_4$ is more complicated with a field-dependent radius of the pinning potential, which is possible, for example, in a collective pinning theory when the vortex lattice evolves from the single-vortex pining regime to the vortex bundle regime [58]. In addition, it is known that the coherence length increases with the magnetic field [60]. Therefore, if $\xi \sim H$, then $\lambda_C$ will be a linear function of the applied field. As for the difference between pristine and irradiated states, it is possible that the collective pinning in the pristine state is replaced by the disordered vortex phase after electron irradiation, and one cannot directly compare the critical current densities using the same formula. In any case, the nature of pinning in $AuSn_4$ requires further investigation.

## 4 Conclusions

We report measurements of London, $\lambda_L(T)$, and Campbell, $\lambda_C(T)$, penetration depths in single crystals of the topological superconductor candidate $AuSn_4$ to elucidate the nature of superconductivity in the bulk. Several independent parameters studied before and after 2.5 MeV electron irradiation unambiguously point to isotropic single $s-$wave gap weak coupling BCS superconductor. Specifically, the superfluid density before and after electron irradiation overlaps almost perfectly with the parameter-free theoretical BCS curves in the full temperature range for clean and dirty limits, respectively. The Campbell penetration depth before and after electron irradiation does not show hysteresis between the ZFC and FC data, indicating a parabolic shape of the pinning potential. However, the $H-$linear behavior of $\lambda_C$ implies either the field-dependent Labusch parameter, $\alpha$, or the radius of the pinning potential, $r_p$, or both. Considering the low pinning in $AuSn_4$ single crystals and the point-like nature of the induced defects, such a field dependence may be expected in the vortex bundle regimes within the collective pinning theory [58].

## Acknowledgments

We thank Hermann Suderow for fruitful discussions.

**Funding information** This work was supported by the US DOE, Office of Science, BES Materials Science and Engineering Division under the contract # DE-AC02-07CH11358. The authors acknowledge support from the EMIRA French network (FR CNRS 3618) on the SIRIUS platform.

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
