# Peer review of "Single-gap Isotropic $s-$wave Superconductivity in Single Crystals $\text{AuSn}_4$"

_SciPost Physics, doi:SciPost Phys. 17, 116 (2024)_

## Round 1 · Referee Report · Anonymous (Referee 1) · 2024-7-21

Report

This work reports a study of the magnetic penetration depth in AuSn4, a candidate topological superconductor, using a tunnel-diode resonator. The results indicate AuSn4 to be a conventional superconductor in the weak coupling limit, and the vortices have certain properties beyond simple models. This work provides unequivocal experimental determination of a critical physical property (the pairing symmetry) in an interesting material, and should be of interest to researchers working on topological superconductors.

I believe this work satisfies 'Open a new pathway in an existing or a new research direction, with clear potential for multi-pronged follow-up work', as it suggests that any nontrivial superconducting behaviors must be limited to the surface, motivating follow-up works along this line. I therefore recommend this work for publication, once the authors address the issues/questions below:

(1) The measurement and analysis should be further described in the Methods section, in particular how Delta lambda_L and Delta_lambda_m are collected (DC vs AC fields), and then how lambda_m is obtained.

(2) In the analysis of Delta lambda in Fig. 1, how would the addition of a small constant affect the analysis (At^n+C rather than just At^n)? This small constant may result from the difference between lambda(T=0) and lambda(0.4K), as in this work Delta lambda = lambda(T)-lambda(0.4K).

(3) the value of delta=Delta(0)/Tc is assumed, can it be independently obtained from the data? How reliable is lambda(0) obtained with this assumption?

(4) Line 141 stated the data are adjusted to match a field-independent value. How are they adjusted (shifted?), and how is the field-independent value obtained (which field?)?

(5) line 163 states 'our estimate of the critical current density is much closer to the true critical value.', what are the estimated values, and what are the 'true' values?

Recommendation

Ask for minor revision

  • validity: high
  • significance: good
  • originality: ok
  • clarity: good
  • formatting: excellent
  • grammar: excellent

Author:  Ruslan Prozorov  on 2024-07-25  [id 4656]

(in reply to Report 1 on 2024-07-21)
Category:
answer to question

1) We have modified the Methods section significantly, explaining the details of the experiment and data analysis. 2) Since, the temperature dependence of the λ(T) is exponential at low temperature, the difference between λ(T=0) and λ(T=0.4K) is much less than the experimental error. We tried to fit with a finite shift parameter “C” but it did not change the result. 3) The BCS ratio, Δ(0)/Tc=1.76, best fits the data when we consider both the λ(T), Fig.2(a), and the full-temperature range superfluid density ρ(T), Fig.2(b). If we let the ratio vary when fitting λ(T), the full-range ρ(T) does not fit. The direct fitting of the superfluid density in the whole temperature range is not simple since the self-consistency equations should be solved. 4) This is described above in our answer to (1), where we have added a detailed procedure in two new paragraphs in the Methods section (paragraphs 3 and 4). 5) We revised this part and no longer use the somewhat ambiguous term “true current density”. Instead, we now write about “theoretical current density”, which is estimated from the shape of the pinning potential well.

---

## Round 1 · Referee Report · Anonymous (Referee 2) · 2024-8-24

Report

This manuscript reports penetration depth measurements for AuSn4. It is a high-quality experimental work that clearly deserves to be published. However, I do not see how it meets the acceptance criteria of SciPost Physics. The authors indicated that their work "details a ground-breaking discovery" and "presents a breakthrough on a previously identified and long-standing research stumbling block". The former does not apply here because, technically, no discovery has been made. The fact of superconductivity of AuSn4 is well-known. The second criterion does not seem to be applicable either, because this work does not demonstrate any breakthrough. The authors infer that "AuSn4 is a robust isotropic s-wave superconductor in the bulk. However, we cannot exclude the possibility that it could have a different type of superconductivity in the surface atomic layers". That's a very agreeable statement, but it basically repeats the findings of the Nature Comm. (2023) publication, which reported that "2D unconventional SC in AuSn4 originates from the mixture of p-wave surface and s-wave bulk contributions".

The first reviewer puts forward a somewhat different view and suggests that this work "opens a new pathway in an existing research direction" by restricting any nontrivial superconductivity to the surface. On the other hand, the conventional nature of superconductivity in the bulk of AuSn4 has been reported in several previous studies, which are cited as Refs. 11 and 13. The conclusion on the s-wave superconductivity in the bulk of AuSn4 is by all means not new. The present work supports and reinforces the existing understanding of superconductivity in AuSn4, but it does not change this understanding. It does not shed light on the possible p-wave pairing at the surface, which is arguably the most interesting feature of the material. Therefore, I believe that this manuscript does not meet the acceptance criteria of SciPost Physics. It should be published in a more specialized journal.

Recommendation

Accept in alternative Journal (see Report)

  • validity: high
  • significance: ok
  • originality: ok
  • clarity: high
  • formatting: excellent
  • grammar: excellent

Author:  Ruslan Prozorov  on 2024-08-29  [id 4725]

(in reply to Report 2 on 2024-08-24)
Category:
remark
answer to question
reply to objection
pointer to related literature

We appreciate the candid feedback provided by the Referee.

Discussing the novelty of the results, let us recall literally thousands of papers on the d-wave superconductivity in the cuprates or two-gap superconductivity in MgB2, all coming to the same conclusion, repeated over and over, yet considered important and worthy of publishing in high-impact journals. Another example, directly relevant here, is multiple experimental studies showing conventional s-wave BCS superconductivity in non-centrosymmetric superconductors that are predicted to have various unconventional properties. We believe that our experimental proof of a conventional pairing state in an alleged topological material with complex bulk vs surface superconductivity is not less important.

On a technical note, our measurements are novel for AuSn4, therefore they add new knowledge. Specifically, there are two new results: the superfluid density in the entire temperature range fitting the predictions of weak-coupling s-wave superconductivity and the validity of the Anderson theorem in response to a non-magnetic disorder produced by electron irradiation. Both experiments are non-trivial and are far from routine. At the end of the day, the objective information can only be reached by adding more and more independent experiments and they should be treated on equal footing.

---

## Round 2 · Referee Report · Anonymous (Referee 3) · 2024-9-27

Report

This manuscript reports penetration depth data in topological superconductor candidate AuSn4. The authors find that bulk AuSn4 is a robust s-wave superconductor. The manuscript has previously been evaluated by two reviewers. All reviewers, including this reviewer, agree that the data quality is superior and the interpretation that bulk AuSn4 is an s-wave superconductor is valid. However, Reviewer #1 and Reviewer #2 disagreed on the impact of the manuscript, leading to opposite recommendations.

Penetration depth measured with tunnel diode resonator is one of the most reliable methods to extract the superconducting gap. In this respect, this work does provide important data for understanding the superconductivity of AuSn4. Furthermore, as far as I am aware the effect of non-magnetic impurities introduced by electron irradiation is discussed for the first time, and the impurity effect is consistent with the s-wave gap. What would be interesting is to revisit the STM work using the irradiated samples. If this follow-up experiment can be designed, this work does ‘open a new pathway in an existing or a new research direction’. Finally, I also compared the two versions – version 2 has been expanded to address Reviewer #1’s comments and hence contains more information. Therefore, I support the publication of this manuscript.

Minor comments:
(a) L199: “Fig. 4 shows a practically linear temperature dependence….”. It should be field dependence instead
(b) L219: The introduction of ‘Labusch’ parameter in Conclusions is abrupt. This term is not used in other parts of the manuscript.

Recommendation

Publish (meets expectations and criteria for this Journal)

---

## Round 2 · Author Response

REFEREE 2

This work reports a study of the magnetic penetration depth in AuSn4, a candidate topological superconductor, using a tunnel-diode resonator. The results indicate AuSn4 to be a conventional superconductor in the weak coupling limit, and the vortices have certain properties beyond simple models. This work provides unequivocal experimental determination of a critical physical property (the pairing symmetry) in an interesting material, and should be of interest to researchers working on topological superconductors. I believe this work satisfies 'Open a new pathway in an existing or a new research direction, with clear potential for multi-pronged follow-up work', as it suggests that any nontrivial superconducting behaviors must be limited to the surface, motivating follow-up works along this line. I therefore recommend this work for publication, once the authors address the issues/questions below.

RESPONSE: We thank the referee for this positive comment and conclusion.

1) The measurement and analysis should be further described in the Methods section, in particular how Delta lambda_L and Delta_lambda_m are collected (DC vs AC fields), and then how lambda_m is obtained.

RESPONSE: We have significantly modified and expanded the Methods section, including more details of the experiment and data analysis.

2) In the analysis of Delta lambda in Fig. 1, how would the addition of a small constant affect the analysis (At^n+C rather than just At^n)? This small constant may result from the difference between lambda(T=0) and lambda(0.4K), as in this work Delta lambda = lambda(T)-lambda(0.4K).

RESPONSE: Since the temperature dependence of the λ(T) is exponential at low temperature, the difference between λ(T=0) and λ(T=0.4K) is much less than the experimental error. We tried to fit with a finite shift parameter “C,” but it did not change the result. 3) the value of delta=Delta(0)/Tc is assumed, can it be independently obtained from the data? How reliable is lambda(0) obtained with this assumption?

RESPONSE: The BCS ratio, Δ(0)/Tc=1.76, best fits the data when we consider both the λ(T), Fig.2(a), and the full-temperature range superfluid density ρ(T), Fig.2(b). If we let the ratio vary, when fitting λ(T), the full-range ρ(T) does not fit. The direct fitting of the superfluid density in the whole temperature range is not simple since the self-consistency equations should be solved. It is worth mentioning that STM measurements found a distribution of gap values and varying surface Tc, most likely due to polytipism of AuSn4 (Hererra et al PRM,2023). The BCS gap ratio of 1.76 is well within the experimentally determined range. 4) Line 141 stated the data are adjusted to match a field-independent value. How are they adjusted (shifted?), and how is the field-independent value obtained (which field?)?

RESPONSE: This is described above in our answer to (1), where we have added detailed procedure in two new paragraphs in the Methods section (paragraphs 3 and 4).

5) Line 163 states 'our estimate of the critical current density is much closer to the true critical value.', what are the estimated values, and what are the 'true' values?

RESPONSE: We revised this part and no longer use the somewhat ambiguous term “true current density”. Instead, we now write about “theoretical current density”, which is estimated from the shape of the pinning potential well.

REFEREE 2

This manuscript reports penetration depth measurements for AuSn4. It is a high-quality experimental work that clearly deserves to be published. However, I do not see how it meets the acceptance criteria of SciPost Physics. The authors indicated that their work "details a ground-breaking discovery" and "presents a breakthrough on a previously identified and long-standing research stumbling block". The former does not apply here because, technically, no discovery has been made. The fact of superconductivity of AuSn4 is well-known. The second criterion does not seem to be applicable either, because this work does not demonstrate any breakthrough. The authors infer that "AuSn4 is a robust isotropic s-wave superconductor in the bulk. However, we cannot exclude the possibility that it could have a different type of superconductivity in the surface atomic layers". That's a very agreeable statement, but it basically repeats the findings of the Nature Comm. (2023) publication, which reported that "2D unconventional SC in AuSn4 originates from the mixture of p-wave surface and s-wave bulk contributions". The first reviewer puts forward a somewhat different view and suggests that this work "opens a new pathway in an existing research direction" by restricting any nontrivial superconductivity to the surface. On the other hand, the conventional nature of superconductivity in the bulk of AuSn4 has been reported in several previous studies, which are cited as Refs. 11 and 13. The conclusion on the s-wave superconductivity in the bulk of AuSn4 is by all means not new. The present work supports and reinforces the existing understanding of superconductivity in AuSn4, but it does not change this understanding. It does not shed light on the possible p-wave pairing at the surface, which is arguably the most interesting feature of the material. Therefore, I believe that this manuscript does not meet the acceptance criteria of SciPost Physics. It should be published in a more specialized journal.

RESPONSE: We appreciate the candid feedback provided by the Referee.

Discussing the novelty of the results, let us recall literally thousands of papers on the d-wave superconductivity in the cuprates or two-gap superconductivity in MgB2, all coming to the same conclusion, repeated over and over, yet considered important and worthy of publishing in high-impact journals. Another example, directly relevant here, is multiple experimental studies showing conventional s-wave BCS superconductivity in non-centrosymmetric superconductors that are predicted to have various unconventional properties. We believe that our experimental proof of a conventional pairing state in an alleged topological material with complex bulk vs surface superconductivity is not less important.

On a technical note, our measurements are novel for AuSn4, adding new knowledge. Specifically, there are two new results: the superfluid density in the entire temperature range fitting the predictions of weak-coupling s-wave superconductivity and the validity of the Anderson theorem in response to a non-magnetic disorder produced by electron irradiation. Both experiments are non-trivial and are far from routine. These two quantities, the superfluid density and superconducting transition temperature, are the two most basic properties. An experimental demonstration of quantitative compliance with the theory is required to prove any suggested pairing state. In this case it happened to be a weak-coupling s-wave BCS, but there are many examples, including from our group, when such analysis shows a different type of the order parameter.

---

## Round 2 · List of Changes

1) We have significantly modified the Methods section significantly, explaining the details of the experiment and data analysis. 2) Since, the temperature dependence of the λ(T) is exponential at low temperature, the difference between λ(T=0) and λ(T=0.4K) is much less than the experimental error. We tried to fit with a finite shift parameter “C” but it did not change the result. 3) The BCS ratio, Δ(0)/Tc=1.76, best fits the data when we consider both the λ(T), Fig.2(a), and the full-temperature range superfluid density ρ(T), Fig.2(b). If we let the ratio vary when fitting λ(T), the full-range ρ(T) does not fit. The direct fitting of the superfluid density in the whole temperature range is not simple since the self-consistency equations should be solved. 4) This is described above in our answer to (1), where we have added a detailed procedure in two new paragraphs in the Methods section (paragraphs 3 and 4). 5) We revised this part and no longer use the somewhat ambiguous term “true current density”. Instead, we now write about “theoretical current density”, which is estimated from the shape of the pinning potential well.

---

## Editorial Decision

published